# Predictors of Turkish individuals' online shopping adoption: An empirical study on regional difference

Şeyda Ünver[1¤a], Ahmet Fatih Aydemir[2¤b], Ömer Alkan[1,3¤a]*

1 Department of Econometrics, Ataturk University, Erzurum, Türkiye, 2 Department of International Trade and Logistics, Ataturk University, Erzurum, Türkiye, 3 Master Araştırma Eğitim ve Danışmanlık Hizmetleri Ltd. Şti., Ata Teknokent, Erzurum, Türkiye

¤a Current address: Department of Econometrics, Faculty of Economics and Administrative Sciences, Ataturk University, Erzurum, Türkiye
¤b Current address: Department of International Trade and Logistics, Faculty of Economics and Administrative Sciences, Ataturk University, Erzurum, Türkiye
* oalkan@atauni.edu.tr

**Citation:** Ünver Ş, Aydemir AF, Alkan Ö (2023) Predictors of Turkish individuals' online shopping adoption: An empirical study on regional difference. PLoS ONE 18(7): e0288835. https://doi.org/10.1371/journal.pone.0288835

**Data Availability Statement:** The data underlying this study is subject to third-party restrictions by the Turkey Statistical Institute. Data are available from the Turkish Statistical Institute (bilgi@tuik. gov.tr) for researchers who meet the criteria for

## Abstract

E-commerce is a result of the most recent technological advancements which has liberated trade on a global scale and further facilitated the communication of information. The purpose of this study is to research the differences in the usage of e-commerce among individuals living in regions of different levels of development in Türkiye and to determine the relationship between demographic, economic and personal characteristics and the e-commerce usage of individuals. In this study, the micro data set obtained from the Household Information Technologies Use Survey was used. Binary logistic regression analysis was used to determine the factors related to the e-commerce usage of individuals. As a result of the study, it was detected that the variables; education level, level of income, age, gender, occupation, social media use, engagement in information search about products and services online, engagement in selling products or services online, use of online banking, and e-government use are related to e-commerce usage. The study found that the variables affecting e-commerce usage differ by region. It is necessary to expand the internet by improving the information and communication technology infrastructure in low development regions to facilitate the development the use of e-commerce in these regions. Therefore, when looking for information on e-commerce, relevant regional initiatives should be considered. In developing countries, expanding e-commerce to low development areas requires more than just facilitating access to the internet.

## Introduction

In the last few centuries, humanity has gone through industrial and electronic revolutions; in the 21st century, the network revolution has taken place. The usage of internet, especially e-commerce, is the driving force behind this network revolution [1]. The Internet plays a vital

access to confidential data. The authors of the study did not receive any special privileges in accessing the data.

**Funding:** The authors received no specific funding for this work.

**Competing interests:** The authors have declared that no competing interests exist.

role in our lives in that it allows people to easily access our world and opens international borders [2]. Internet technology advancements have changed the way of commerce and transformed business models into electronic commerce (e-commerce) [3]. New businesses and commercial instruments have been developed as a result of advances in information technology, which are effective means of facilitating new arrangements between businesses and clients. The earliest examples of electronic commerce involve electronic money transfer. Eventually, effective transfer transactions between financial institutions followed [4]. E-commerce is a product of the most recent technological advancements observed in recent years, which facilitates the communication of information, as well as the liberalization of trade on a global scale [5].

E-commerce is the execution, processing, and facilitation of commercial activities over computer networks [6]. E-commerce has become one of the largest megatrends in the global economy, with a very comprehensive scope [7]. Different forms of e-commerce include online shopping, electronic payments, online ticketing, and internet banking [8, 9]. Online shopping is one of the most prevalent forms of Internet-based activities. Online shopping, which refers to the purchase and sale of goods over the Internet, is a form of producer-to-consumer e-commerce [10]. It is possible to reach any market or consumer through rapid communication due to the vast scope of e-commerce activities, which range from individuals to other businesses, the government, and other related organizations [11]. E-commerce activities may therefore be economic, political, legal, institutional, cultural, social, etc. It is influenced by many different factors that can be defined in various classifications [12]. As the use of e-commerce enhances opportunities in a variety of fields, it is crucial to evaluate the magnitude of differences and their underlying causes [13].

Online shopping saves consumers both time and travel costs. In addition, online stores enable consumers to shop 24 hours a day [11]. Online shopping also allows consumers to make the best choice among the many options available [14]. Online dealers have the advantage of reaching customers all over the world at a low cost as they do not have to invest substantial amounts of money to establish physical stores. Reduced seller expenses are reflected as discounted prices on products sold to the consumer [6]. These are a few reasons as to why online shopping has become so popular in a very short time.

Economic and social development on Earth, varies over time and space [15]. This difference takes advantageous spaces forward but leaves others behind. Therefore, spatial inequality paves the way for the occurrence of this problem. This problem can take place between countries as well as between regions of a country. Today, the regional development disparity applies to all countries, whether developed or developing, and thus requires countries to make a great effort to overcome this problem [16].

The digital gap between regions with different levels of development affects how telecommunications and other advanced technologies are used [17]. Socioeconomic factors affect the use of information and communication technology and also create regional disparities [18]. The spread of the Internet occurs at the intersection of international and domestic socioeconomic inequalities [19]. Consequently, socioeconomic status plays a significant role in determining how people integrate online shopping into their daily lives [20]. Access to resources and financial conditions affect the extent of people's involvement in online shopping [21, 22]. The differences in the infrastructure, economy and population between regions cause the environmental diversification of locations [23] thus, also affecting the difference of citizens' online shopping proclivities by region [24, 25].

In recent years, the rapid development of online shopping has necessitated careful identification of the primary factors influencing consumer behavior and attitudes toward online shopping [26]. Individuals utilize e-commerce in a variety of methods and for a variety of purposes

[27]. The distribution of different demographic groups is worth analyzing in terms of the use of e-commerce by regions. This study performs a systematic analysis to investigate the impact of various selected factors on e-commerce usage among individuals by region. Since regional development is a significant factor in the growth of individuals' e-commerce usage, it is evident that the effect of the level of development on e-commerce usage is becoming increasingly significant [28].

Internet was first introduced to Türkiye in 1993. People's intentions for using the internet have changed with each passing day since it was first made available for home use in 1996 [29]. A new economic order has emerged with the fast development of internet use and communication technologies in the world. Parallel to this, geographical boundaries have been removed: The markets where buyers and sellers meet have expanded and sellers now recognize customers from all over the world and have shifted their business activities to web-based systems [30].

E-commerce began to materialize substantially after the 1990s and has been on the rise since the early 2000s [31]. Türkiye exhibits a significant growth potential in e-commerce, with an increasing number of e-dealers and a large youth population. According to the Informatics Industry Association of Türkiye (TÜBİSAD), e-commerce grew by 42% in Türkiye in 2018 [32]. In a study that encompasses 28 countries, including European countries and Türkiye, the average internet usage of people who have ordered or purchased a product or service over the internet in the last year is 87% in Europe, 72% in Türkiye; In the same research, while the average of the people who ordered or purchased a product or service in the last year was 60% in Europe, it was calculated as 25% in Türkiye [33].

Little is known about regional differences of e-commerce usage in Türkiye. To the best of our knowledge, this is the first known study conducted to determine the factors that are related to the regional usage of e-commerce in Türkiye. In this study, the following research questions were focused on regarding the regional usage of e-commerce in Türkiye:

Research Question 1: Is the e-commerce usage of individuals different by region?

Research Question 2: Is there a relationship between demographic, economic and personal characteristics of different regions and the e-commerce usage of individuals?

Research Question 3: Are the factors related to the e-commerce usage of individuals in regions with different levels of development the same?

## Literature review

Online shopping is becoming a dominant alternative to traditional retail shopping [34]. As a result of online shopping, consumers can search for more information and choose to compare the pricing of products, providing them with more options and convenience. Moreover, online shopping facilitates higher consumer satisfaction and saves consumers time [35]. The first literature on e-commerce was primarily concerned with determining how e-commerce affects the price levels and distribution [36]. According to the earliest empirical findings, online markets do not have lower price distribution than traditional markets [37]. Later empirical research, on the other hand, demonstrates that online markets tend to have a lower price distribution than traditional markets [36].

Academicians have performed extensive research to learn more about consumer behavior on e-commerce platforms. In addition, consumer- and technology-oriented studies have attempted to provide information on consumer behavior from various viewpoints. Consumer-oriented studies evaluated consumer shopping behaviors from the various demographical perspectives of consumers [36, 38], entailing cultural factors [39], psychological factors [40, 41], trust factors [42–44] and risk perception [45–48]. In a study conducted on undergraduate and graduate students in Taiwan, it was stated that consumers' trust in online retailers affects not

only their attitudes towards online shopping, but also their intentions for online shopping [44].

On the other hand, technology-oriented studies evaluated consumer behavior from the perspective of online privacy and security [41, 49–52], technical features of stores such as design [39, 49, 53], ease of navigation [49] and information content [44, 49]. The studies found that online consumers' privacy and security concerns are the main factors that significantly influence their online purchase intentions [41, 49–52].

In a study conducted in Ethiopia, the factors affecting the feasibility and readiness of e-commerce in developing countries were investigated [54]. Another study investigated the function of TQMk (Total Quality Marketing) in enhancing the efficacy of e-marketing in the Jordanian telecommunications industry. The study suggests focusing on consumers in target markets from multiple perspectives, including price, new product acceptance, customer behavior, and motivations for purchasing decisions [55]. On the other hand, according to a study, word-of-mouth (WOM) is regarded as the most important and comprehensive communication channel for consumers who desire more information and wish to disseminate their information [56]. According to a previous study, WOM, in addition to conveying knowledge and experience regarding the products/services of an e-commerce retailer, also contributes to the formation of recommendations regarding the use of the e-commerce site itself. [57]. Some previous studies have also stated that with a positive word-of-mouth marketing intention, customers are more likely to share positive experiences and opinions about an e-commerce website and recommend it to others through interpersonal communication [58, 59].

It was found that factors such as age, level of income and education level have a substantial impact on the online buying behaviors of public and private sector employees in Türkiye [60]. The effects of gender, age and level of income on individuals' internet use was researched in another study conducted concerning public and private sector employees [61]. In a study conducted in Ankara, the attitudes of university students who use the internet towards online shopping were investigated. Financial challenges, product quality issues, refunding issues, product delivery issues, security and privacy concerns have all been found to be critical factors in online shopping concerns [6].

The use of e-commerce in Türkiye has also been the subject of extensive research. In recent years, researchers in Türkiye have identified factors that influence individuals' online purchasing behavior [6, 26, 28, 60–63]. In the study, which reveals visually the spatial distribution of e-commerce utilization rates in Türkiye by province, it was determined that there is a heterogeneous spatial distribution across Türkiye. In areas with a small population, e-commerce consumption rates have been observed to be skewed [64]. It has been determined that the factors of age, gender, level of education, occupation, level of income, region, and household size affect the use of e-commerce by individuals in Türkiye [63]. In addition, the use of e-commerce varies according to the education level of individuals in Türkiye, and that the propensity towards e-commerce increases as education level rises [26]. As a result of another study conducted with households in Türkiye, it was determined that factors such as education, age, marital status, employment status, income, life insurance ownership, credit card usage, automobile ownership, and the year of the survey affect households' online shopping behavior [65]. Working women with a higher level of education, who reside in urban areas, and who have smaller families are more likely to use the internet and purchase supplies online. Similarly, while age was found to have a significant impact on internet usage, it had no effect on online grocery shopping [66]. In a study investigating sociodemographic factors affecting online purchasing behavior in Türkiye, it was found that age, income and education level had a significant effect on online purchasing behavior [60]. In the study that determined socioeconomic and demographic factors that affect the purchase or ordering of products or services over the Internet in

Türkiye, it was determined that women are more likely than men to shop online for clothing and sports equipment [62]. According to a study, women are more likely to engage in e-commerce, but their e-commerce expenditures are lower than those of men [67]. In another study, it was found that anxiety has a greater impact on the consumer intentions of older women with less online purchasing experience [68]. It has been determined that demographic variables such as age, gender, income, and level of education have a substantial impact on consumers' online purchasing frequency. It has been stated that gender has a significant effect on what consumers purchase, how frequently they purchase, and why they purchase particular products/services [69].

There are also studies that examine the utilization of e-commerce in Türkiye from various angles. There are studies that emphasize e-satisfaction, e-loyalty, and e-service quality, particularly in relation to online purchasing [31, 70, 71]. Additionally, there are studies examining the relationship between e-commerce and business logistics [29, 72]. In another study, a new approach was introduced to extract customers' shopping patterns from mouse movements and website logs. As a result of the study, it was determined that the product adding time and shopping time are crucial for women to purchase the products added to their shopping carts [73]. In another study, a Secure E-commerce Scheme (SES) is proposed that reduces communication costs for all parties while mitigating security threats posed by e-commerce companies. The secure e-commerce protocol SES proposed in this study was compared with two well-known Secure Electronic Transactions (SET) and 3D Secure schemes [74].

## Materials and methods

### Data

This study was accomplished by using data of the survey titled the Household Information Technologies Use Survey conducted by Turkish Statistical Institute. Therefore, ethical approval was not required for this study.

We used secondary data for this study. In order to use the micro dataset from the Household Information Technologies Use Survey, the official permission was obtained from the Turkish Statistical Institute. In addition, a "Letter of Undertaking" was given to the Turkish Statistical Institute for the use of the data subjected to the study.

In this study, the micro data set of the Household Information Technologies (IT) Use Survey conducted in 2019 by the Turkish Statistical Institute was used. The Household Information Technologies Use Survey has been being conducted since 2004 to collect data about the use of information and communication technologies owned by households and individuals, and is the main source of such data. In the Household Information Technologies Usage Survey, every settlement in Türkiye was included in the sample selection. Households in all settlements within Türkiye's borders are considered covered. The institutional population, which is comprised of those living in schools, dormitories, hotels, kindergartens, nursing homes, hospitals, and prisons, as well as those living in barracks and army residences, is not included. In addition, settlements whose population is estimated to be less than 1% of the total population (small villages, obas, hamlets, etc.) and where the sample household count cannot be reached are excluded from the scope. Individuals between the ages of 16–74 were included in the survey as the methodology of the study required. The sampling method of the research is 2-stage stratified cluster sampling. In the first stage, clusters (blocks) containing an average of 100 households were selected with probability proportional to their size (PPS), and in the second stage, sample addresses were determined using the systematic selection method from the clusters selected for the sample. The Classification of Statistical Regional Units is used as a Level 1 stratification criterion. It is designed to generate estimates for the 16–74 age group on the basis

of the totals of Türkiye and the Statistical Regional Units Classification Level 1 (NUTS Level1) totals. Since the selection probabilities are used due to the multi-stage sample design from the data set obtained as a result of the sampling, the weighting process is performed. The final weight to be determined is a combination of many factors. Thus, the estimated population of individuals aged 16 to 74 (excluding the institutional population) and the total number of households in Türkiye are reached [75].

Computer-Assisted Personal Interview Method (CAPI) was used to collect data until 2020, and Computer-Assisted Telephone Interview Method (CATI) was used after 2020. The research questionnaire is based on a model questionnaire developed and recommended by the European Union Statistical Office (EUROSTAT). This questionnaire is adapted to the conditions of Türkiye and is also prepared in line with the needs of the institutions/organizations that implement the National Information Society Strategy. TUIK confirmed that informed consent was obtained from all participants for research [75].

In this study, the data from a total of 28,675 participants from the 2019 Household Information Technologies Use Survey was used, including 10,821 from the western region, 9,852 from the central region and 8,002 from the eastern region.

## Outcome variables

At the basis of employing the Nomenclature of Units for Territorial Statistics (NUTS) in Türkiye lies the obligation to establish Development Agencies. Türkiye is divided into 12 regions in Level 1 under the Nomenclature of Territorial Units for Statistics (NUTS). In the study, these regions are categorized into the western, central and eastern regions. These regions and the provinces of these regions are shown in detail in Table 1 [76].

The dependent variable of the study is the e-commerce usage of the individuals from the western, central and eastern regions in the last year. The participants were given the code "1" if they had used e-commerce within the last year, and "0" if they had not at the time the survey was conducted. An approximate logit model was configured separately for each region in the study.

## Independent variables

The independent variables included in this study are the variables that exist in the Household Information Technologies Use Survey and are found as a result of the literary research. The quantitative independent variables of the study are the number of IT equipment in the

**Table 1. Nomenclature of territorial units for statistics—Level 1.**

| Region | Code | Level 1 | Provinces |
|---|---|---|---|
| Western Region | TR1 | İstanbul | İstanbul |
| | TR2 | West Marmara | Tekirdağ, Edirne, Kırklareli, Balıkesir, Çanakkale |
| | TR3 | Aegean | İzmir, Aydın, Denizli, Muğla, Manisa, Afyonkarahisar, Kütahya, Uşak |
| | TR4 | East Marmara | Bursa, Eskişehir, Bilecik, Kocaeli, Sakarya, Düzce, Bolu, Yalova |
| Central Region | TR5 | Western Anatolia | Ankara, Konya, Karaman |
| | TR6 | Mediterranean | Antalya, Isparta, Burdur, Adana, Mersin, Hatay, Kahramanmaraş, Osmaniye |
| | TR7 | Central Anatolia | Kırıkkale, Aksaray, Niğde, Nevşehir, Kırşehir, Kayseri, Sivas, Yozgat |
| | TR8 | West Blacksea | Zonguldak, Karabük, Bartın, Kastamonu, Çankırı, Sinop, Samsun, Tokat, Çorum, Amasya |
| Eastern Region | TR9 | East Blacksea | Trabzon, Ordu, Giresun, Rize, Artvin, Gümüşhane |
| | TRA | NortheasternAnatolia | Erzurum, Erzincan, Bayburt, Ağrı, Kars, Iğdır, Ardahan |
| | TRB | East Anatolia | Malatya, Elâzığ, Bingöl, Tunceli, Van, Muş, Bitlis, Hakkâri |
| | TRC | Southeastern Anatolia | Gaziantep, Adıyaman, Kilis, Şanlıurfa, Diyarbakır, Mardin, Batman, Şırnak, Siirt |

household and the household size. The categorical variables included in the model were measured using nominal and ordinal scales. The qualitative independent variables are; level of income, age, gender, education level, occupation, social media use in the last three months, engagement in information search about products and services online in the last three months, engagement in selling products or services online in the last three months, online banking use, and e-government use in the last 12 months. Ordinal and nominal variables were defined as dummy variables with the goal to observe the effects of categories of all variables to be included in the binary logistic regression model [77].

## Research model and hypotheses

**Research model.** In the study, demographic factors constitute the independent (decision) variables. In literature, various researchers have conducted studies which suggest that demographic factors affect the attitudes of individuals towards online buying from different viewpoints [30, 42, 78, 79]. In a study conducted in South Africa, it was found that demographic variables such as level of income, education and age may have a significant effect on the purchase choices of consumers [80]. Various studies have found that demographic characteristics play a significant role in customer decision-making, for example by forming customer perceptions and attitudes towards different marketing strategies [81, 82]. However, no study has dealt with demographic factors differing particularly by region. In this study, the effect of various selected factors on e-commerce usage among individuals by region was investigated.

The independent and dependent variables of the study are associated according to the research model shown in Fig 1. The variables in the research model are assumed as follows.

## Hypotheses related to demographic factors

People use e-commerce in different ways and for various reasons [27]. Given that e-commerce usage creates better opportunities in many areas, it is important to evaluate the magnitude of differences and their underlying causes [13]. Therefore, the differences in e-commerce use in

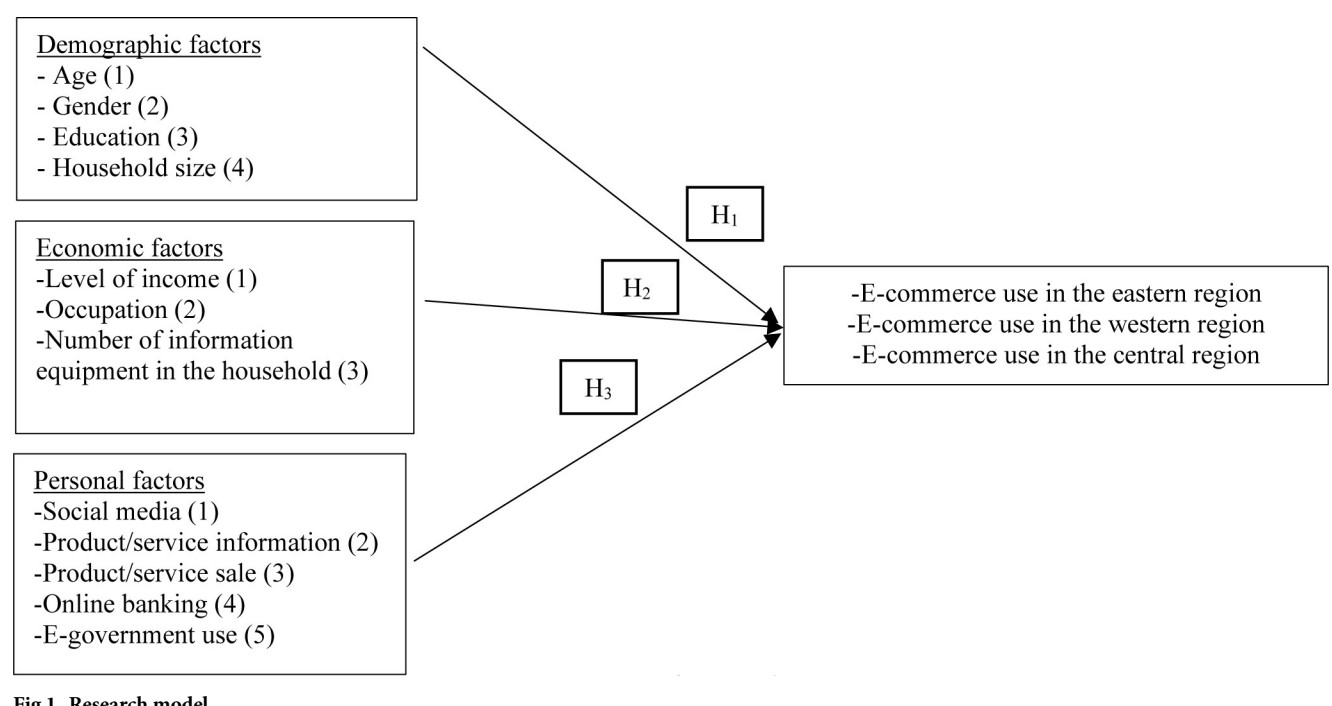

**Fig 1. Research model.**

various respects between demographic groups is the research topic [83]. It was found that demographic characteristics of individuals affect their behavior [84]. In the literature, studies have been conducted by various researchers showing that demographic factors affect individuals' attitudes towards online purchasing behavior [30, 42, 78, 79]. Thus, the following hypotheses are offered:

$H_{11}$: *There is a significant relationship between age and e-commerce usage.*

$H_{21}$: *There is a significant relationship between gender and e-commerce usage.*

$H_{31}$: *There is a significant relationship between education level and e-commerce usage.*

$H_{41}$: *There is a significant relationship between household size and e-commerce usage.*

## Hypotheses related to economic factors

E-commerce is defined in general as an economic activity that is performed over an electronic connection [85]. Economic factors that affect e-commerce usage include; level of income, occupation, and number of IT equipment in the household [86, 87]. The data obtained by the US National Telecommunications and Information Administration (NTIA) shows that economic factors are associated with e-commerce usage [88]. In a study in which it was stated that effects on consumer behavior generally occur between external and internal factors, it was found that the improvement of external socioeconomic conditions had a significant impact on e-commerce usage [89]. Thus, the following hypotheses are offered:

$H_{12}$: *There is a significant relationship between level of income and e-commerce usage.*

$H_{22}$: *There is a significant relationship between occupation and e-commerce usage.*

$H_{32}$: *There is a significant relationship between number of IT equipment in the household and e-commerce usage.*

## Hypotheses related to personal factors

In literature, there is evidence indicating that personal factors affect the attitudes of individuals towards e-commerce usage [30, 42, 78, 79]. According to the Theory of Planned Behavior (TPB), personal factors are some of the most important factors that influence the intended and actual behaviors of individuals [90]. Likewise, a number of authors have considered this approach to explain online shopping behavior [4]. Limayem, Khalifa [91] observed in their study on the adoption of e-commerce, that purchase intention and perceived control directly influenced effective behaviour [91]. In various studies, it was found that personal factors related to e-commerce usage significantly affect online shopping intentions and individual behaviours [40, 52, 92]. Hence:

$H_{13}$: *There is a significant relationship between social media use and e-commerce usage.*

$H_{23}$: *There is a significant relationship between engaging in information search about products and services and e-commerce usage.*

$H_{33}$: *There is a significant relationship between engaging in selling products or services and e-commerce usage.*

$H_{43}$: *There is a significant relationship between online banking use and e-commerce usage.*

$H_{53}$: *There is a significant relationship between e-government use and e-commerce usage.*

## Statistical analysis

One of the main areas of statistical inference is the testing of statistical hypotheses. SPSS 20 and Stata 15 software were used to analyze the data. Firstly, the frequencies and percentages of the study participants for demographic, economic and personal factors were obtained by regions. In this study, the binary logistic regression method was used to research the relationship between demographic, economic and personal factors and e-commerce usage by regions. Binary logistic regression is a statistical analysis method used to examine the causal relationship between the dependent variable and the independent variable(s) when the dependent variable has two possible states [93, 94].

# Results

## Characteristics of study participants

As of the time when the survey was conducted, it was found that 35.32% of individuals living in the western region, 30.29% of individuals living in the central region and 16.55% of individuals living in the eastern region made purchases online during the last year. Findings on the factors that are related to the e-commerce usage of individuals in Türkiye by region are shown in Table 2.

## Model estimation

The binary logistic regression model was used to determine the factors related to the e-commerce usage of the participants by region. Estimated model results are shown in Table 3. Whether there is multicollinearity between the independent variables to be included in the binary logistic regression model was tested. It is considered that those with variance inflation factor (VIF) values of 5 and above cause multicollinearity at a moderate degree, and those with 10 and above cause multicollinearity at a high degree [95]. There is no variable in this study that causes a multicollinearity problem between the variables.

Table 3 shows that the variables; level of income, age, gender, education level (secondary school, high school, university), occupation (executives, professional staff, plant-machinery operators/installers, unskilled workers), social media use, product/service information, product/service sale, use of online banking, e-government use, number of IT equipment in the household and household size were significant in the model established for the western region of Türkiye. The variables; level of income (₺6001 and more), age, gender, education level (secondary school, high school, university), occupation (executives, professionals, technicians and associate professionals, service/sales personnel, skilled agriculture/forestry/aquaculture workers, craftsman/similar workers, plant-machinery operators/installers, unskilled workers), social media use, product/service information, product/service sale, use of online banking, e-government use, number of IT equipment in the household, and household size were significant in the model established for the central region of Türkiye. Additionally, Table 3 shows that the variables; level of income (₺2001-₺4000, ₺6001 and above), age (35–44, 45–54, 55–64, 65 and older), gender, education level, occupation (unskilled workers), social media use, product/service information, product/service sale, use of online banking, e-government use, number of IT equipment in the household and household size were significant in the model established for the eastern region of Türkiye.

Marginal effects of demographic, economic and personal factors related to the e-commerce usage of individuals are shown in Table 4.

According to the binary logistic regression model shown in Table 4, an individual with an income of ₺6001 and above in the western region was 36.6% more likely to use e-commerce

**Table 2. Findings on factors related to e-commerce use by region.**

| Qualitative Variables | | Western Region (n = 10821) | | Central Region (n = 9852) | | Eastern Region (n = 8002) | |
|---|---|---|---|---|---|---|---|
| | | N | % | n | % | n | % |
| **Level of income** | ₺2000 and less | 2,534 | 23.4 | 3,138 | 31.9 | 3,973 | 49.7 |
| | ₺2001–₺4000 | 4,662 | 43.1 | 3,922 | 39.8 | 2,770 | 34.6 |
| | ₺4001–₺6000 | 2,217 | 20.5 | 1,693 | 17.2 | 763 | 9.5 |
| | ₺6001 and more | 1,408 | 13.0 | 1,099 | 11.2 | 496 | 6.2 |
| **Age** | 15–24 | 1,428 | 13.2 | 1,467 | 14.9 | 1,615 | 20.2 |
| | 25–34 | 2,108 | 19.5 | 1,833 | 18.6 | 1,621 | 20.3 |
| | 35–44 | 2,517 | 23.3 | 2,133 | 21.7 | 1,618 | 20.2 |
| | 45–54 | 2,050 | 18.9 | 1,869 | 19 | 1,355 | 16.9 |
| | 55–64 | 1,717 | 15.9 | 1,542 | 15.7 | 1,096 | 13.7 |
| | 65 and older | 1,001 | 9.3 | 1,008 | 10.2 | 697 | 8.7 |
| **Gender** | Male | 5,579 | 51.6 | 5,127 | 52 | 4,240 | 53 |
| | Female | 5,242 | 48.4 | 4,725 | 48 | 3,762 | 47 |
| **Education level** | Uneducated | 840 | 7.8 | 1,036 | 10.5 | 2,027 | 25.3 |
| | Primary School | 3,650 | 33.7 | 3,198 | 32.5 | 2,191 | 27.4 |
| | Secondary School | 1,777 | 16.4 | 1,743 | 17.7 | 1,569 | 19.6 |
| | High School | 2,308 | 21.3 | 2,030 | 20.6 | 1,214 | 15.2 |
| | University | 2,246 | 20.8 | 1,845 | 18.7 | 1,001 | 12.5 |
| **Occupation** | Executives | 199 | 1.8 | 155 | 1.6 | 41 | 0.5 |
| | Professional staff | 721 | 6.7 | 734 | 7.5 | 382 | 4.8 |
| | Technicians/assistant professional staff | 128 | 1.2 | 94 | 1 | 40 | 0.5 |
| | Office staff | 476 | 4.4 | 464 | 4.7 | 232 | 2.9 |
| | Service/sales personnel | 1,016 | 9.4 | 740 | 7.5 | 534 | 6.7 |
| | Skilled agriculture/forestry/aquaculture workers | 231 | 2.1 | 509 | 5.2 | 321 | 4 |
| | Craftsman/similar workers | 319 | 2.9 | 324 | 3.3 | 126 | 1.6 |
| | Plant and machine operators/installers | 316 | 2.9 | 192 | 1.9 | 34 | 0.4 |
| | Unskilled workers | 1,480 | 13.7 | 1,175 | 11.9 | 1,137 | 14.2 |
| | Unemployed | 5,935 | 54.8 | 5,465 | 55.5 | 5,155 | 64.4 |
| **Social media** | Yes | 6,997 | 64.7 | 5,593 | 56.8 | 3,652 | 45.6 |
| | No | 3,824 | 35.3 | 4,259 | 43.2 | 4,350 | 54.4 |
| **Product/service information** | Yes | 5,749 | 53.1 | 4,625 | 46.9 | 2,532 | 31.6 |
| | No | 5,072 | 46.9 | 5,227 | 53.1 | 5,470 | 68.4 |
| **Product/service sale** | Yes | 1,845 | 17.1 | 1,446 | 14.7 | 1,066 | 13.3 |
| | No | 8,976 | 82.9 | 8,406 | 85.3 | 6,936 | 86.7 |
| **Online banking** | Yes | 4,250 | 39.3 | 3,256 | 33 | 1,636 | 20.4 |
| | No | 6,571 | 60.7 | 6,596 | 67 | 6,366 | 79.6 |
| **E-government use** | Yes | 5,977 | 55.2 | 4,941 | 50.2 | 2,731 | 34.1 |
| | No | 4,844 | 44.8 | 4,911 | 49.8 | 5,271 | 65.9 |
| **Quantitative Variables** | | Western Region | | Central Region | | Eastern Region | |
| | | Mean | S. Dev. | Mean | S. Dev. | Mean | S. Dev. |
| **Number of information equipment in the household** | | 2.46 | 1.269 | 2.27 | 1.2 | 1.82 | 1.053 |
| **Household size** | | 3.44 | 1.461 | 3.64 | 1.6 | 4.85 | 2.407 |

than the reference group (₺2000 and less). An individual in the age group of 25–34 and living in the western region was 25.2% less likely to use e-commerce than the reference group (15–24). Similarly, an individual in the age group of 65 and older was 165% less likely to use e-

**Table 3. Estimated model results of demographic, economic and personal factors related to e-commerce use of individuals.**

| Variables | Western Region | | Central Region | | Eastern Region | |
|---|---|---|---|---|---|---|
| | B | S.E | β | S.E | β | S.E |
| **Constant** | -3.146[a] | 0.320 | -3.189[a] | 0.344 | -4.157[a] | 0.346 |
| **Level of income (reference category: ₺2000 and less)** | | | | | | |
| ₺2001-₺4000 | 0.245[b] | 0.099 | 0.063 | 0.098 | 0.369[a] | 0.124 |
| ₺4001-₺6000 | 0.425[a] | 0.113 | 0.168 | 0.115 | -0.026 | 0.180 |
| ₺6001 and more | 0.564[a] | 0.130 | 0.487[a] | 0.144 | 0.660[a] | 0.194 |
| **Age (reference category: 15–24)** | | | | | | |
| 25–34 | -0.439[a] | 0.103 | -0.330[a] | 0.106 | 0.087 | 0.148 |
| 35–44 | -0.847[a] | 0.104 | -0.815[a] | 0.115 | -0.285[c] | 0.154 |
| 45–54 | -1.530[a] | 0.118 | -1.511[a] | 0.132 | -1.067[a] | 0.179 |
| 55–64 | -2.187[a] | 0.146 | -2.072[a] | 0.160 | -1.984[a] | 0.287 |
| 65 and older | -2.374[a] | 0.226 | -2.300[a] | 0.306 | -1.879[a] | 0.361 |
| **Gender (reference category: male)** | | | | | | |
| Female | -0.348[a] | 0.072 | -0.368[a] | 0.083 | -0.578[a] | 0.124 |
| **Education level (reference category: uneducated)** | | | | | | |
| Primary School | 0.289 | 0.308 | 0.300 | 0.3220 | 0.830[a] | 0.323 |
| Secondary School | 0.662[b] | 0.310 | 0.808[b] | 0.324 | 1.055[a] | 0.333 |
| High School | 0.943[a] | 0.309 | 0.856[a] | 0.322 | 1.492[a] | 0.331 |
| University | 1.310[a] | 0.316 | 1.305[a] | 0.329 | 1.782[a] | 0.342 |
| **Occupation (reference category: unemployed)** | | | | | | |
| Executives | 0.734[a] | 0.245 | 0.507[b] | 0.252 | -0.550 | 0.458 |
| Professional staff | 0.328[b] | 0.146 | 0.365[b] | 0.155 | -0.108 | 0.231 |
| Technicians/assistant professional staff | 0.268 | 0.287 | 0.765[b] | 0.312 | 0.348 | 0.392 |
| Office staff | 0.190 | 0.153 | 0.123 | 0.149 | 0.289 | 0.255 |
| Service/sales personnel | 0.044 | 0.108 | -0.230[c] | 0.135 | -0.052 | 0.183 |
| Skilled agriculture/forestry/aquaculture workers | -0.356 | 0.331 | -1.050[a] | 0.305 | -0.400 | 0.397 |
| Craftsman/similar workers | -0.271 | 0.180 | -0.638[a] | 0.195 | -0.295 | 0.317 |
| Plant and machine operators/installers | -0.394[b] | 0.180 | -0.672[a] | 0.230 | -0.529 | 0.716 |
| Unskilled workers | -0,437[a] | 0.109 | -0.427[a] | 0.131 | -0.617[a] | 0.165 |
| **Social media (reference category: no)** | | | | | | |
| Yes | 0.564[a] | 0.091 | 0.652[a] | 0.093 | 0.759[a] | 0.139 |
| **Product/service information (reference category: no)** | | | | | | |
| Yes | 0.943[a] | 0.081 | 0.875[a] | 0.084 | 0.708[a] | 0.120 |
| **Product/service sale (reference category: no)** | | | | | | |
| Yes | 0.764[a] | 0.079 | 0.980[a] | 0.090 | 1.209[a] | 0.120 |
| **Online banking (reference category: no)** | | | | | | |
| Yes | 1.179[a] | 0.078 | 1.188[a] | 0.091 | 1.091[a] | 0.131 |
| **E-government use (reference category: no)** | | | | | | |
| Yes | 0.599[a] | 0.088 | 0.852[a] | 0.102 | 0.674[a] | 0.135 |
| **Number of information equipment** | 0.238[a] | 0.029 | 0.195[a] | 0.033 | 0.267[a] | 0.048 |
| **Household size** | -0.105[a] | 0.026 | -0.095[a] | 0.026 | -0.124[a] | 0.028 |

[a] $p < .01$
[b] $p < .05$
[c] $p < .10$

**Table 4. Marginal effects of demographic, economic and personal factors related to e-commerce use of individuals.**

| Variables | Western Region | | Central Region | | Eastern Region | |
|---|---|---|---|---|---|---|
| | ME | S.E | ME | S.E | ME | S.E |
| **Level of income (reference category: ₺2000 and less)** | | | | | | |
| ₺2001-₺4000 | 0.164[b] | 0.067 | 0.044 | 0.069 | 0.309[a] | 0.104 |
| ₺4001-₺6000 | 0.280[a] | 0.075 | 0.118 | 0.081 | -0.022 | 0.153 |
| ₺6001 and more | 0.366[a] | 0.085 | 0.334[a] | 0.098 | 0.545[a] | 0.158 |
| **Age (reference category: 15–24)** | | | | | | |
| 25–34 | -0.252[a] | 0.058 | -0.211[a] | 0.068 | 0.070 | 0.120 |
| 35–44 | -0.509[a] | 0.061 | -0.543[a] | 0.076 | -0.235[c] | 0.127 |
| 45–54 | -0.986[a] | 0.077 | -1.066[a] | 0.095 | -0.912[a] | 0.154 |
| 55–64 | -1.496[a] | 0.108 | -1.522[a] | 0.125 | -1.750[a] | 0.264 |
| 65 and older | -1.650[a] | 0.181 | -1.715[a] | 0.257 | -1.652[a] | 0.333 |
| **Gender (reference category: male)** | | | | | | |
| Female | -0.224[a] | 0.046 | -0.256[a] | 0.058 | -0.480[a] | 0.102 |
| **Education level (reference category: uneducated)** | | | | | | |
| Primary School | 0.212 | 0.231 | 0.232 | 0.252 | 0.743[b] | 0.295 |
| Secondary School | 0.471[b] | 0.232 | 0.604[b] | 0.255 | 0.936[a] | 0.304 |
| High School | 0.654[a] | 0.231 | 0.637[b] | 0.253 | 1.301[a] | 0.301 |
| University | 0.877[a] | 0.234 | 0.939[a] | 0.257 | 1.534[a] | 0.308 |
| **Occupation (reference category: unemployed)** | | | | | | |
| Executives | 0.439[a] | 0.137 | 0.334[b] | 0.160 | -0.466 | 0.396 |
| Professional staff | 0.204[b] | 0.089 | 0.243[b] | 0.102 | -0.090 | 0.192 |
| Technicians/assistant professional staff | 0.168 | 0.176 | 0.493[a] | 0.189 | 0.283 | 0.314 |
| Office staff | 0.120 | 0.095 | 0.084 | 0.101 | 0.236 | 0.206 |
| Service/sales personnel | 0.028 | 0.069 | -0.161[c] | 0.096 | -0.043 | 0.152 |
| Skilled agriculture/forestry/aquaculture workers | -0.237 | 0.226 | -0.780[a] | 0.242 | -0.337 | 0.339 |
| Craftsman/similar workers | -0.179 | 0.121 | -0.460[a] | 0.146 | -0.247 | 0.268 |
| Plant and machine operators/installers | -0.262[b] | 0.124 | -0.486[a] | 0.173 | -0.447 | 0.619 |
| Unskilled workers | -0.293[a] | 0.074 | -0.303[a] | 0.094 | -0.524[a] | 0.142 |
| **Social media (reference category: no)** | | | | | | |
| Yes | 0.378[a] | 0.063 | 0.469[a] | 0.069 | 0.645[a] | 0.120 |
| **Product/service information (reference category: no)** | | | | | | |
| Yes | 0.637[a] | 0.058 | 0.628[a] | 0.062 | 0.597[a] | 0.102 |
| **Product/service sale (reference category: no)** | | | | | | |
| Yes | 0.476[a] | 0.047 | 0.653[a] | 0.057 | 0.982[a] | 0.094 |
| **Online banking (reference category: no)** | | | | | | |
| Yes | 0.775[a] | 0.052 | 0.829[a] | 0.064 | 0.902[a] | 0.107 |
| **E-government use (reference category: no)** | | | | | | |
| Yes | 0.400[a] | 0.060 | 0.616[a] | 0.077 | 0.568[a] | 0.115 |
| **Number of information equipment** | 0.154[a] | 0.019 | 0.136[a] | 0.023 | 0.223[a] | 0.040 |
| **Household size** | -0.068[a] | 0.017 | -0.066[a] | 0.019 | -0.104[a] | 0.023 |

[a] $p < .01$

[b] $p < .05$

[c] $p < .10$

commerce than the reference group. A woman in the western region was 22.4% less likely to use e-commerce than a man. An individual with a university degree in the western region was 87.7% more likely to use e-commerce than the reference group (having no university diploma). An individual who was an executive living in the western region was 43.9% more likely to use e-commerce than the reference group (unemployed), while an individual who was an unskilled worker living in the same region was 29.3% less likely to use e-commerce than the reference group. In the western region, an individual who had engaged in information search about products and services online in the last three months was 63.7% more likely to use e-commerce than others.

An individual with an income of ₺6001 and above in the central region was 33.4% more likely to use e-commerce than the reference group. An individual in the age group of 25–34 and living in the central region was 21.1% less likely to use e-commerce than the reference group. Similarly, an individual in the age group of 65 and older was 171.5% less likely to use e-commerce than the reference group. A woman in the central region was 25.6% less likely to use e-commerce than a man. A university graduate in the central region was 93.9% more likely to use e-commerce than the reference group. An individual who was a technicians and associate professionals in the central region was 49.3% more likely to use e-commerce than the reference group. An individual who was a skilled agriculture/forestry/aquaculture worker was 78% less likely to use e-commerce than the reference group.

An individual with an income of ₺6001 and above in the eastern region was 54.5% more likely to use e-commerce than the reference group. An individual in the age group of 35–44 and living in the eastern region was 23.5% less likely to use e-commerce than the reference group. Similarly, an individual in the age group of 55–64 was 175% less likely to use e-commerce than the reference group. A woman in the eastern region was 48% less likely to use e-commerce than a man. An individual with a university degree in the eastern region was 153.4% more likely to use e-commerce than the reference group. An individual who was an unskilled worker in the eastern region was 52.4% less likely to use e-commerce than the reference group.

Individuals in the eastern region, who used social media, had engaged in online product or service sale in the last three months, used online banking and had used an e-government application in the last 12 months were 64.5%, 98.2%, 90.2% and 56.8% more likely to use e-commerce than others. 1% increase in the number of IT equipment in the household increased the likelihood of e-commerce usage in the eastern region by 22.3%. 1% increase in household size decreases the likelihood of e-commerce usage in the eastern region by 10.4%.

## Discussion

The ubiquitous nature of the Internet enables the majority of modern information technology systems to provide services on a global scale [96]. As online users take advantage of the convenience of global services, international data transfer and e-commerce become crucial factors to consider. In recent years, the use of e-commerce and online purchasing has increased due to the expanding reach and adoption of new connected mobile and social applications. It is crucial that both e-commerce providers and online shoppers understand the factors associated with online purchasing [97]. Regional development, on the other hand, is an essential foundation for the growth of online consumption, and the level of development has a growing impact on online consumption [10].

In this study, the data from a total of 28,675 participants from the 2019 Household Information Technologies Use Survey was used, including 10,821 from the western region, 9,852 from the central region and 8,002 from the eastern region.

The study concluded that as the education level of an individual increases, the likelihood of them using e-commerce increases. Other studies have also reached similar conclusions [26, 60,

87]. Education is an important demographic factor that affects the decision-making process of customers while online shopping. Customers with different education levels have different expectations and thus different perceived service quality value [98]. Customers with a higher level of education demand not only a broad range of products but also a higher level of service [99]; they want to be served by professional staff. As for professional services, the perceived value of customers with a higher education level is different from that of customers with a lower education level [14]; as the education level of individuals increases, their tendency towards online shopping increases. Higher education level implies higher level of income in both public and private sectors and higher perception of innovations. Naturally, this has a positive effect on the amount of online shopping individuals engage in [60]. According to the findings of a similar study conducted in Thailand, the majority of online shoppers are university students [100]. In another study examining the effect of education level on online shopping habits, it was found that university students are more influenced by family, friends and social media [101].

As the level of income of individuals increases, their likelihood of e-commerce use increases. Other studies have also reached similar conclusions [28, 61, 86]. This can be explained by the fact that individuals with higher level of income are likely to have higher qualifications and new technologies [61]. Furthermore, consumers with different levels of income have different perceptions of service quality [102]. Customers with a higher level of income expect higher quality service interactions [103]. Therefore, customers with a high level of income are more likely to pay higher prices for better services [14]. On the other hand, according to a study conducted in New Zealand, the higher a person's income, the greater their trust in online sales [104]. In another study examining the factors affecting e-commerce motivation, it was found that income level and economic conditions have a significant impact on the online purchasing behavior of individuals living in Greece, particularly their price and promotion sensitivity [10].

Age is one of the factors that affect individuals' attitudes toward e-commerce use, and various age groups may have different e-commerce usage tendencies [86]. It was determined in the study that as the age of individuals increases, the likelihood of e-commerce use decreases. Parallel results have been found in different studies [60, 80, 105]. It is argued that shopping online requires more physical effort as consumers grow old [105]. Customers who are older than 30 years have different decision-making processes and habits than younger customers. Customers in this age group place a higher value on their time more than younger customers, and older customers are more likely to link their sacrifices in terms of time, money and effort to the perceived value of a service provision [106, 107]. On the other hand, there are studies which found that as individuals get older, their likelihood of using e-commerce increases [104, 108]. In the studies, this is explained by the level of income of the participants. In those studies, young people are in the low-income group, and this negatively affects their attitudes towards internet use. Moreover, the studies have shown that older individuals have more opportunities to use the internet as they have more spare time [61]. In a study conducted in New Zealand, it was determined that the amount of time spent by the elderly on Internet e-commerce was comparable to that of other age groups [104].

According to the study, men were more likely to use e-commerce than women. Other studies have also arrived at similar conclusions [27, 84, 86]. In various studies, the reasons for men to prefer online shopping more compared to women were stated as time efficiency, avoiding crowds and the ability to shop 24 hours a day [5, 109]. Moreover, some studies found significant evidence that women exhibit lower levels of computer skills and higher levels of computer anxiety [27, 83, 105]. In addition, women do not trust online shopping and do not embrace online shopping as much as men [110]. In a study conducted in Türkiye, it was found that the

effect of the gender variable differs by product groups. According to the study, the preference for shopping of foodstuffs, daily necessities and household goods does not differ in terms of gender. Also, it was stated in the study that women shop online for clothing and sports equipment more than men [62]. On the other hand, some studies found that the gender variable does not affect e-commerce usage. Studies have attributed this result to the fact that participants with similar backgrounds and positions within their society may be more likely to exhibit similar attitudes and behaviors in the use of ICT [60, 84, 104].

In the central region, it was detected that technicians/assistant professionals use e-commerce more than unemployed individuals, while skilled agriculture/forestry/aquaculture workers use e-commerce less than unemployed individuals. In the western region, it was determined that executives use e-commerce more than unemployed individuals, while plant-machinery operators/installers use e-commerce less than unemployed individuals.

This can be explained by the fact that the professional characteristics of individuals in different societies can change depending on their different positions. Other studies have also reached similar conclusions [60, 61, 111].

It was determined that individuals who use social media use e-commerce more than those who do not. Individuals who manage social media accounts are more involved in buying through online channels. Parallel results have been found in different studies [34, 112]. Online dealers offer their products or services to potential consumers through social media platforms such as Facebook, Instagram, Twitter etc. [34]. It was also detected that individuals who are engaged in searching information about products and services online use e-commerce more than others. Other studies have also reached similar conclusions [113, 114].

It was found that individuals who are engaged in selling products or services online use e-commerce more than others. Parallel results have been found in different studies [26, 115, 116]. Emergence of various online markets and auction sites enables individuals to make online sales very easily [116, 117]. Another important factor affecting e-commerce usage is online social networks. Social networks are a platform where buyers can share their experiences about e-commerce. The most popular online auction sites are available on these networks and advertise through them [115, 118].

Internet banking users engage in e-commerce more frequently than other individuals. Other studies have also arrived at similar conclusions [34, 119]. The increase in internet access and the development of internet financing have led to a dramatic increase in internet-based purchases [34]. It has been determined that individuals who use e-government services engage in e-commerce more frequently than others [61, 120].

The likelihood of using e-commerce increases as the number of IT devices grows Other studies have also arrived at similar conclusion [62, 121, 122]. The study found that the likelihood of using e-commerce decreases as household size increases [28, 62, 123]. In contrast to this research, another study found that the likelihood of using e-commerce increases with the size of the household [124]. On the other hand, studies have found that the household size variable does not affect e-commerce usage [125].

## Theoretical and practical implications

In order to fill the relevant gap in the literature, it is believed that conducting the study on a large sample size and incorporating demographic factors, which are not typically included in studies conducted in this context, will be crucial. In addition, since the study has a methodological perspective and is conducted using data collection and analysis methods based on modeling, it is possible to determine the personal and environmental factors influencing individuals' use of e-commerce, to have more usage options, to keep the duration of use longer, to

increase sectoral occupational groups, to diversify demographic characteristics, and to determine the relationships between these factors. It allows for the realization of various scenarios.

## Conclusion

In the study, using binary logistic regression analysis, the factors affecting the use of e-commerce by individuals according to regions in Türkiye were determined. According to the results of the analysis, the variables of education level, income level, age, gender, occupation, use of social media, searching for information about goods and services on the internet, selling goods and services on the internet, internet banking use, and e-government use were found to be associated with e-commerce.

Our findings indicate that it is necessary to facilitate the growth of e-commerce in regions with low levels of development and to increase Internet usage by enhancing the infrastructure of information and communication technologies in these regions. Therefore, region-specific interventions should be considered when gaining access to e-commerce information. The results of the study can provide important information to academics and policymakers on how to encourage the use of e-commerce in developing countries in order to increase the volume of e-commerce shopping in Türkiye, the operational efficiency of e-commerce applications, customer satisfaction, and social welfare and quality of life. Our data set contains the variables that were considered for the investigation. In future research, variables such as internet access in the household, internet consumption time, and household members who can make internet purchases can be incorporated into the analysis.

Türkiye's share of e-commerce is increasing day by day, increasing its trade volume and potential in both national and international markets, making significant returns to the country's economy. Despite the e-transformation rates observed in the public and private sectors, the increasing number of e-commerce users, legal and sectoral regulations, it is seen that the e-commerce potential cannot be fully evaluated and that developed countries are lagging behind. For this purpose, the first steps to be taken are the development of technological infrastructure and standardization. Despite high urbanization rates, regional disparities in Internet access rates remain a significant problem. Eliminating regional disparities, increasing investments in fixed and mobile infrastructures, reducing taxes on products and services to make ICT technologies accessible to all, and encouraging and assisting new entrepreneurs to enter the market can be cited as the most crucial steps for the development of e-commerce. In addition, the development of legal regulations, the revitalization of the entrepreneurial ecosystem, and the increase in investments in techno-parks will both stimulate the domestic market and attract foreign investments to the country, resulting in micro and macroeconomic growth and a significant increase in Türkiye's competitiveness.

This study, like almost all other studies, has a few limitations. First, the data in this study are secondary data and the variables required for statistical analysis consist of existing variables in the data set. In addition, since the data are cross-sectional, the definite causal relationship between e-commerce usage and related socio-economic factors cannot be inferred [126]. Finally, the data collected for this study are the participants' own responses. Therefore, the data obtained through this method of data collection may be biased.

## Acknowledgments

The authors would like to thank the Turkish Statistical Institute for the data. The views and opinions expressed in this manuscript are those of the authors only and do not necessarily represent the views, official policy, or position of the Turkish Statistical Institute.

## Author Contributions

**Conceptualization:** Ömer Alkan.

**Data curation:** Şeyda Ünver.

**Formal analysis:** Şeyda Ünver, Ömer Alkan.

**Investigation:** Şeyda Ünver.

**Methodology:** Ömer Alkan.

**Validation:** Ahmet Fatih Aydemir.

**Writing – original draft:** Şeyda Ünver, Ahmet Fatih Aydemir, Ömer Alkan.

**Writing – review & editing:** Şeyda Ünver, Ahmet Fatih Aydemir, Ömer Alkan.

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
