## [Decision Letter · Decision Letter 0]

19 Apr 2023

PONE-D-23-08202Predictors of Turkish Individuals’ Online Shopping Adoption: An Empirical Study on Regional DifferencePLOS ONE

Dear Dr. Alkan,

Thank you for submitting your manuscript to PLOS ONE. After careful consideration, we feel that it has merit but does not fully meet PLOS ONE’s publication criteria as it currently stands. Therefore, we invite you to submit a revised version of the manuscript that addresses the points raised during the review process.

We look forward to receiving your revised manuscript.

Kind regards,

Ahmad Samed Al-Adwan

Academic Editor

PLOS ONE

Journal Requirements:

Additional Editor Comments (if provided):

Thank you for submitting your paper to PLOSE ONE. We believe that your submission has a merit. However, there are serious issues that need to be addressed by you in order to make your submission suitable for publication. All comments are highlighted in the next section. 

The reviewers' comments

- The novelty of this paper is presented in a very limited manner. In the introduction, it is important to emphasize the significance of the work and justify its novelty by highlighting its main contributions to the existing literature.

- The proposed hypotheses need more theoretical support.

- Please provide a description of the population under study and the sampling technique utilized in the methodology and research design section. Additionally, please provide a rationale for the chosen sample size and sampling technique.

- The authors are advised to dedicate a separate section to highlight the main theoretical and practical implications.

- The literature review is presented in a very limited manner. Thus, it is recommended to extend the literature by including further recent, related and well-established studies. This includes but not limited to:

E-commerce in high uncertainty avoidance cultures: The driving forces of repurchase and word-of-mouth intentions. Doi: https://doi.org/10.1016/j.techsoc.2022.102083

Feasibility assessment for E-commerce: A data collection from developing country (Ethiopia). Doi: https://doi.org/10.1016/j.mex.2022.101639

- As e-commerce is highly researched, the discussion part can be broadened a bit more with more relevant research studies.

- As an academic paper, it is important to have a conclusion section that summaries the main objective of the study, main findings, and limitations and future work.

Reviewers' comments:

Reviewer's Responses to Questions

**Comments to the Author**

1. Is the manuscript technically sound, and do the data support the conclusions?

Reviewer #1: Yes

Reviewer #2: Yes

2. Has the statistical analysis been performed appropriately and rigorously? 

Reviewer #1: Yes

Reviewer #2: Yes

3. Have the authors made all data underlying the findings in their manuscript fully available?

Reviewer #1: Yes

Reviewer #2: Yes

4. Is the manuscript presented in an intelligible fashion and written in standard English?

Reviewer #1: Yes

Reviewer #2: Yes

5. Review Comments to the Author

Reviewer #1: The novelty of this paper is presented in a very limited manner.

In the introduction, it is important to emphasize the significance of the work and justify its novelty by highlighting its main contributions to the existing literature.

The proposed hypotheses need more theoretical support.

Please provide a description of the population under study and the sampling technique utilized in the methodology and research design section.

Additionally, please provide a rationale for the chosen sample size and sampling technique.

The authors are advised to dedicate a separate section to highlight the main theoretical and practical implications.

Reviewer #2: • The literature review is presented in a very limited manner. Thus, it is recommended to extend the literature by including further recent, related and well-established studies. This includes but not limited to:

E-commerce in high uncertainty avoidance cultures: The driving forces of repurchase and word-of-mouth intentions. Doi:

https://doi.org/10.1016/j.techsoc.2022.102083

Feasibility assessment for E-commerce: A data collection from developing country (Ethiopia). Doi: https://doi.org/10.1016/j.mex.2022.101639

• As e-commerce is highly researched, the discussion part can be broadened a bit more with more relevant research studies.

• As an academic paper, it is important to have a conclusion section that summaries the main objective of the study, main findings, and limitations and future work.

6. PLOS authors have the option to publish the peer review history of their article (what does this mean?). If published, this will include your full peer review and any attached files.

Reviewer #1: **Yes: **Dr. Husam Yaseen

Reviewer #2: **Yes: **Dr. Malek Alsoud

---

## [Author Response · Author response to Decision Letter 0]

19 Jun 2023

Dear Editor and Reviewers,

Thank you very much for your comments concerning our manuscript titled “Determinants of Intimate Partner Controlling Behaviour Targeting Women in Türkiye”. These comments have been very helpful in reviewing and improving our manuscript. We have carefully revising these instructive comments and made corrections that we hope will be approved. The revised parts are highlighted in red in the main paper.

Additional Editor Comments (if provided):

Thank you for submitting your paper to PLOSE ONE. We believe that your submission has a merit. However, there are serious issues that need to be addressed by you in order to make your submission suitable for publication. All comments are highlighted in the next section. 

The reviewers' comments

Comment: The novelty of this paper is presented in a very limited manner. In the introduction, it is important to emphasize the significance of the work and justify its novelty by highlighting its main contributions to the existing literature.

Response: Thank you for the comment. Taking this criticism into account, we added the novelty of this paper.

Comment: The proposed hypotheses need more theoretical support.

Response: Thank you for the comment. Taking this criticism into account, we added additional explanations for the hypotheses.

Comment: Please provide a description of the population under study and the sampling technique utilized in the methodology and research design section. Additionally, please provide a rationale for the chosen sample size and sampling technique.

Response: Thank you for the comment. Taking this criticism into account, we added the necessary explanations to the Data section. 

Comment: The authors are advised to dedicate a separate section to highlight the main theoretical and practical implications.

Response: Thank you for the comment. Taking this criticism into account, we added the “Theoretical and practical implications” heading.

Comment: The literature review is presented in a very limited manner. Thus, it is recommended to extend the literature by including further recent, related and well-established studies. This includes but not limited to:

E-commerce in high uncertainty avoidance cultures: The driving forces of repurchase and word-of-mouth intentions. Doi: https://doi.org/10.1016/j.techsoc.2022.102083

Feasibility assessment for E-commerce: A data collection from developing country (Ethiopia). Doi: https://doi.org/10.1016/j.mex.2022.101639

Response: Thank you for the comment. Taking this criticism into account, we have cited both the recommended studies and other new studies for literature review. 

Comment: As e-commerce is highly researched, the discussion part can be broadened a bit more with more relevant research studies.

Response: Thank you for the comment. Taking this criticism into account, we revised the discussion section.

Comment: As an academic paper, it is important to have a conclusion section that summaries the main objective of the study, main findings, and limitations and future work.

Response: Thank you for the comment. Taking this criticism into account, we added the “Conclusion” heading.

 Reviewers' comments:

Reviewer #1: 

Comment: The novelty of this paper is presented in a very limited manner. In the introduction, it is important to emphasize the significance of the work and justify its novelty by highlighting its main contributions to the existing literature.

Response: Thank you for the comment. Taking this criticism into account, we added the novelty of this paper.

Comment: The proposed hypotheses need more theoretical support.

Response: Thank you for the comment. Taking this criticism into account, we added additional explanations for the hypotheses.

Comment: Please provide a description of the population under study and the sampling technique utilized in the methodology and research design section. Additionally, please provide a rationale for the chosen sample size and sampling technique.

Response: Thank you for the comment. Taking this criticism into account, we added the necessary explanations to the Data section. 

Comment: The authors are advised to dedicate a separate section to highlight the main theoretical and practical implications.

Response: Thank you for the comment. Taking this criticism into account, we added the “Theoretical and practical implications” heading.

Reviewer #2: 

Comment: The literature review is presented in a very limited manner. Thus, it is recommended to extend the literature by including further recent, related and well-established studies. This includes but not limited to:

E-commerce in high uncertainty avoidance cultures: The driving forces of repurchase and word-of-mouth intentions. Doi:

https://doi.org/10.1016/j.techsoc.2022.102083

Feasibility assessment for E-commerce: A data collection from developing country (Ethiopia). Doi: https://doi.org/10.1016/j.mex.2022.101639

Response: Thank you for the comment. Taking this criticism into account, we have cited both the recommended studies and other new studies for literature review. 

Comment: As e-commerce is highly researched, the discussion part can be broadened a bit more with more relevant research studies.

Response: Thank you for the comment. Taking this criticism into account, we revised the discussion section.

Comment: As an academic paper, it is important to have a conclusion section that summaries the main objective of the study, main findings, and limitations and future work.

Response: Thank you for the comment. Taking this criticism into account, we added the “Conclusion” heading.

---

## [Editor Report · Decision Letter 1]

6 Jul 2023

Predictors of Turkish Individuals’ Online Shopping Adoption: An Empirical Study on Regional Difference

PONE-D-23-08202R1

Dear Dr. Alkan,

We’re pleased to inform you that your manuscript has been judged scientifically suitable for publication and will be formally accepted for publication once it meets all outstanding technical requirements.

Kind regards,

Ahmad Samed Al-Adwan

Academic Editor

PLOS ONE

Additional Editor Comments (optional):

Thank you for resubmitting the revised version of your paper. The quality of your paper has enhanced significantly.  
---

## [Editor Report · Acceptance letter]

13 Jul 2023

PONE-D-23-08202R1 

Predictors of Turkish Individuals’ Online Shopping Adoption: An Empirical Study on Regional Difference 

Dear Dr. Alkan:

I'm pleased to inform you that your manuscript has been deemed suitable for publication in PLOS ONE. Congratulations! Your manuscript is now with our production department. 

Kind regards, 

on behalf of

Prof. Ahmad Samed Al-Adwan 

Academic Editor

PLOS ONE